# Experimental Study of the Effect of Tack Coats on Interlayer Bond Strength of Pavement

**Mohammed H. Ali** [1], **Amjad H. Khalil** [1,*] and **Yu Wang** [2]

1 Civil Engineering Department, University of Baghdad, Baghdad 10071, Iraq
2 Civil Engineering Department, University of Salford, Manchester M5 4WT, UK
* Correspondence: a.khalil@uobaghdad.edu.iq

**Abstract:** The performance and lifetime of the flexible asphalt pavement are mainly dependent on the interfacial bond strength between layer courses. To enhance the bond between layers, adhesive materials, such as tack coats, are used. The tack coat itself is a bituminous material, which is applied on an existing relatively non-absorbent surface to ensure a strong bond between the old and newly paved layer. The primary objective of this study was to evaluate the effects of various types of tack coat materials on interlayer bond strength and to determine the optimal application rate for each type. The tack coat types used in this paper were RC-70, RC-250, and CSS-1h. Both laboratory-prepared and field-constructed hot mix asphalt concrete pavements using the tack coats were tested for the binding strength between the layers. A direct shear test was used for the testing. The results obtained from the study showed that the optimum application rate for RC-70 was 0.1 L/m$^2$, and for RC-250, it was 0.2 L/m$^2$, while the optimum application rate for CSS-1h was 0.1 L/m$^2$. From the field test, the optimum application rate of the RC-250 tack coat was 0.1 L/m$^2$.

**Keywords:** interlayer bond; pavement; tack coat; bond strength; direct shear test

## 1. Introduction

In general, the hot mix asphalt (HMA) concrete pavement is constructed by layers as the maximum thickness of a single hot mix asphalt layer is limited due to compaction issues [1–3]. The interfacial bonding shear strength between two asphalt concrete layers has, of course, a decisive role in the performance and service life of pavements. The bonding conditions between pavement layers affect the stress, strain, and deflection conditions that develop under loading [4]. After application, around 90% of the pavement surface should be covered with a thin, uniform coating of tack coat material. An appropriate tack coat application rate is important for the obtaining of a high interlayer shear strength between pavement layers. The use of little or no tack coat results in poor structural behavior and premature failure of the pavement structure due to the poor bond between the asphalt concrete layers [5]. The most prevalent distress is the slippage failure, which frequently occurs at the areas of acceleration and deceleration of traffic [6]. The slippage is thought to be the result of the horizontal shear stress due to wheel load exceeding the interlayer bond shear strength [7]. When interfacial de-bonding occurs, the two sides of the HMA pavement deform in separate directions, where the horizontal loads meet no resistance from the slipped layers, resulting in a reduction in the structural bearing capacity of the pavement [8]. Such deteriorations can reduce pavement life by 25–50%, necessitating extensive repairs such as full-depth patches or total reconstruction [9,10]. Moreover, other types of pavement distress, including early fatigue bottom-up cracking and potholes, can occur due to the poor interlayer bond. Conversely, applying excessive tack coat has been reported to result in a lubricated slippage plane between the layers and a decrease in the adhesion and interlocking resistance, which can also cause the tack coat material to be drawn into an overlay, negatively affecting the mix properties and even creating a potential for bleeding in the thin overlays [5].

To provide enough binding strength between the bituminous pavement layers for the layers to function as a single monolithic structure, a tack coat is frequently applied therein. According to ASTM D8, Standard Terminology Relating to Materials for Roads and Pavements, "Tack coat (bond coat) is an application of bituminous material to an existing largely non-absorbent surface to produce a comprehensive bond between old and new surfacing" [9]. The application of a tack coat to asphalt layers has a real impact on the shear strength of the interface bond. This impact is dependent on several variables, including the type of tack coat, the condition of the pavement, and the moisture content [11]. The materials used for tack coats are often hot bituminous binders, cutback bitumen (bitumen–solvent base), and/or bitumen emulsions (bitumen–water base). Compared with cutback asphalt or hot bituminous binders, asphalt emulsions are the most frequently used tack coat materials because of their stability at lower temperatures, their lower environmental impact, and their safety in use due to the fact that they contain no hazardous volatile or flammable solvents [12,13].

Several factors need to be taken into consideration to implement the tack coats; these include the amount of usage, the emulsion type, the regularity of the tack coat application, the roughness of the contacting layers, the cleanliness of the contact surface, the temperature, and the loading [14].

Mohammad et al. [15] also conducted both laboratory and field tests for five different kinds of tack coat materials to find out the optimum application rates, the optimum application methods, and the effective testing procedures to evaluate the bond shear strength between the tack coat interlayer and the different types of sublayer, such as the milled asphalt layer course, new asphalt layer course, aged asphalt layer course, and grooved concrete layer course. It was found that the texture of the underlying layer course has a direct effect on the bond strength. The identified bonding strength order is as follows: the milled layer course, the grooved concrete layer course, and the old asphalt layer course; the new asphalt layer showed the weakest bond with the tack coat. The study also showed that the bonding strength measured in the laboratory was higher than that measured on-site.

Joni et al. [16] investigated the interfacial bond strength between the tack coats and the flexible overlayer course. They used the FDOT test method and studied two tack coat types; they were Anionic CSS-1 and Rc-70. The tack coats were applied at rates of 0.15, 0.25, and 0.35 L/m$^2$ for Anionic CSS-1 and 0.3, 0.4, and 0.5 L/m$^2$ for Rc-70. The results showed that the optimum rate was 0.25 L/m$^2$ for the emulsion (Anionic and CSS-1) and 0.4 L/m$^2$ for Rc-70. Temperature was found to have a considerable effect on the bond strength. The maximum bond strength was found at 15 °C with the use of the tack coat. However, the highest bond strength was at 30 °C for the pavement using no tack coat.

West et al. [8] conducted laboratory and field experimental research to evaluate the interfacial binding effect for the types of tack coat, the application rate (usage), the types of asphalt mixture, and the temperature. Among all the factors, the temperature showed the most pronounced influence; when the temperature increased, the interfacial bonding strength showed a bigger reduction.

Recently, Saad and Abdul Razaaq [6] studied the moisture effect on the interfacial bond strength of the multilayer pavement. They studied two types of tack coats, called RC-70 and CMS. The application rates were 0.15, 0.33, 0.5 L/m$^2$ for RC-70, and 0.1, 0.23, and 0.35 L/m$^2$ for CMS. After being subjected to moisture exposure, the specimens of RC-70 tack coat showed greater permanent deformation and lower shear strength than the CMS specimens at all the application rates. The trend of the results indicates that the interface bond strength decreased under a repeated load with moisture conditions compared to the samples under a repeated load alone.

Another recent work [17] investigated three types of tack coat, namely CRS-2P, CSS-1h, and SS-h, at four rates: 0, 0.14, 0.281, and 0.702 L/m$^2$. The shear strength was tested for four two-layered pavements. The upper layer was hot mix asphalt (HMA), while the lower layers were HMA, aged and worn HMA, milled HMA, and grooved Portland cement

concrete, respectively. The study concluded that the CSS-1h tack coat could be utilized effectively on all types of surfaces.

Wei et al. [18] investigated the use of a tack coat to improve the interfacial bond between an HMA overlay and a PCC underlayer. They studied four types of tack coats: cutback asphalt, anionic emulsified asphalt, rubber asphalt, and virgin asphalt. A direct shear test was conducted at two different temperatures. The results indicated that at 15 °C the emulsified asphalt showed the highest interfacial bond, which was followed by the rubber asphalt, virgin asphalt, and cutback asphalt, respectively. At 45 °C, the virgin asphalt showed the highest bond, followed by cutback asphalt, rubber asphalt, and emulsified asphalt, respectively.

All the previous works primarily focused on the experiments conducted in a laboratory but contained little verification of the results in the field. However, for real world engineering applications, the conditions in the field are much more complex and are not as well controlled as those in a lab. To increase the knowledge of tack coat application under field conditions and to compare the difference between the lab and the field measurements, this paper reports research on three types of tack coat materials to evaluate their effect on interlayer bond strength and to determine their respective optimal application rates. Both the laboratory-prepared and the field-constructed hot mix asphalt concrete pavements using the tack coats were tested for their shear strength.

## 2. Raw Materials

All the materials used in this study follow the specification of the State Corporation for Roads and Bridges (SCRB) [19].

### 2.1. Aggregate

The aggregate was from Al-Nibaee Quarry, the main supplier in the Baghdad area for asphaltic mixtures [20–22]. Five aggregate fractions were used by the mixing plant to produce asphalt concrete, namely coarse aggregate (25–4.75 mm), coarse aggregate (19–4.75 mm), midsize aggregate (12.5–2.36 mm), crusher sand, and natural sand. In the lab, The gradation of the aggregate fractions provided from the material stock of the patch plant was obtained through a complete sieve analysis of each fraction. The gradation of the aggregates and the suggested mixing ratios for the surface and binder layers are shown in Tables 1 and 2. Figures 1 and 2 show the final aggregate gradation for the binder layer and surface layer. The physical properties of the coarse aggregate that was retained in sieve No. 4 (4.75 mm) and those of the fine one that was retained in sieve No. 200 (0.075 mm) are listed in Tables 3–5.

**Table 1.** Aggregate mixing ratio for binder layer.

| Sieve Size, mm (inch) | Percent Passing | | | | | Final Gradation | SCRB Specification Requirements |
|---|---|---|---|---|---|---|---|
| | The Gradation of Aggregate Samples | | | | | | |
| | Coarse Agg. (25–4.75 mm) | Midsize Agg. (12.5–2.36 mm) | Crusher Sand | Natural Sand | Filler | | |
| 25 (1) | 100 | 100 | 100 | 100 | 100 | 100 | 100 |
| 19 (3/4) | 88 | 100 | 100 | 100 | 100 | 96 | 90–100 |
| 12.5 (1/2) | 46 | 100 | 100 | 100 | 100 | 81 | 70–90 |
| 9.5 (3/8) | 9 | 92 | 100 | 100 | 100 | 67 | 56–80 |
| 4.75 (No. 4) | 0 | 10 | 85 | 97 | 100 | 46 | 35–65 |
| 2.36 (No. 8) | 0 | 0 | 60 | 83 | 100 | 34 | 23–49 |
| 0.30 (No. 50) | 0 | 0 | 3 | 51 | 100 | 10 | 5–19 |
| 0.075 (No. 200) | 0 | 0 | 1 | 6 | 95 | 6 | 3–9 |
| Mixing Ratio | 35% | 15% | 37% | 8% | 5% | | |

**Table 2.** Aggregate mixing ratio for surface layer.

| Sieve Size, mm (inch) | Coarse Agg. (19–4.75 mm) | Midsize Agg. (12.5–2.36 mm) | Crusher Sand | Natural Sand | Filler | Final Gradation | SCRB Specification Requirements |
|---|---|---|---|---|---|---|---|
| | **Percent Passing** | | | | | | |
| | **The Gradation of Aggregate Samples** | | | | | | |
| 19 (3/4) | 100 | 100 | 100 | 100 | 100 | 100 | 100 |
| 12.5 (1/2) | 60 | 100 | 100 | 100 | 100 | 94 | 90–100 |
| 9.5 (3/8) | 10 | 92 | 100 | 100 | 100 | 84 | 76–90 |
| 4.75 (No. 4) | 0 | 10 | 85 | 97 | 100 | 54 | 44–74 |
| 2.36 (No. 8) | 0 | 0 | 60 | 83 | 100 | 39 | 28–58 |
| 0.30 (No. 50) | 0 | 0 | 3 | 51 | 100 | 11 | 5–21 |
| 0.075 (No. 200) | 0 | 0 | 1 | 6 | 95 | 6 | 4–10 |
| Mixing Ratio | 15% | 27% | 43% | 10% | 5% | | |

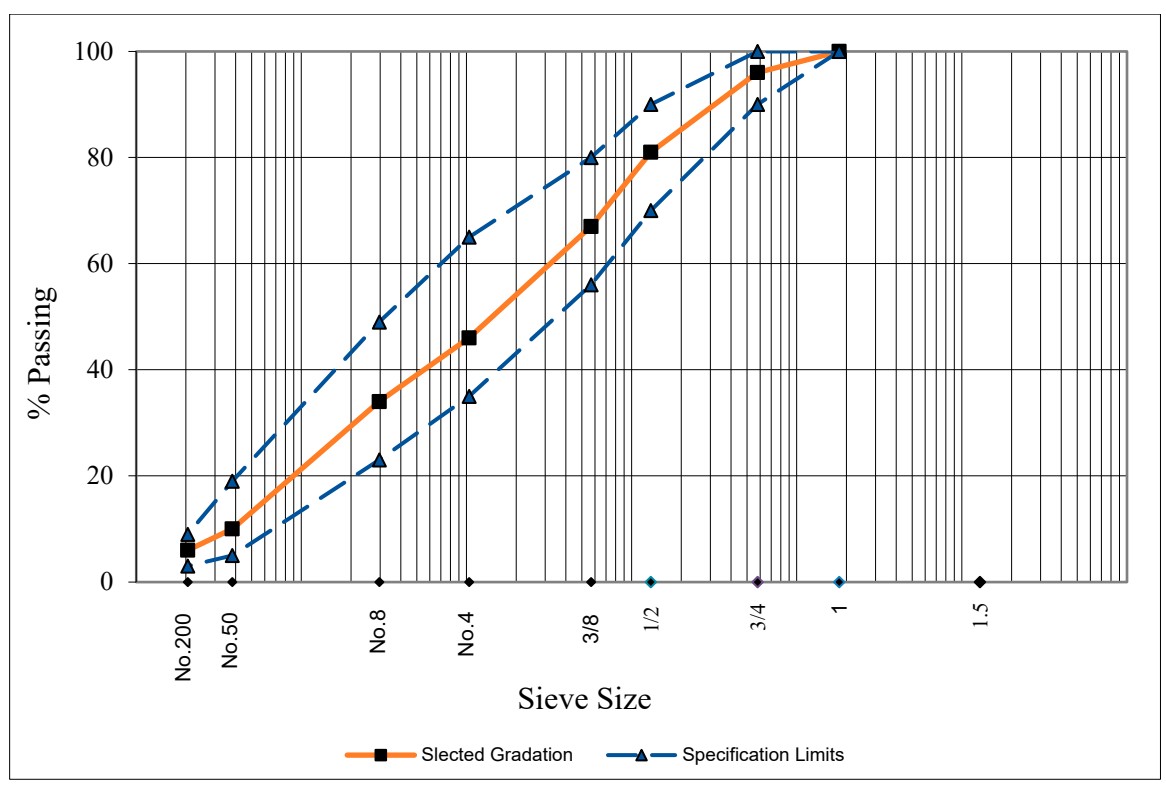

**Figure 1.** The aggregate gradation for binder course.

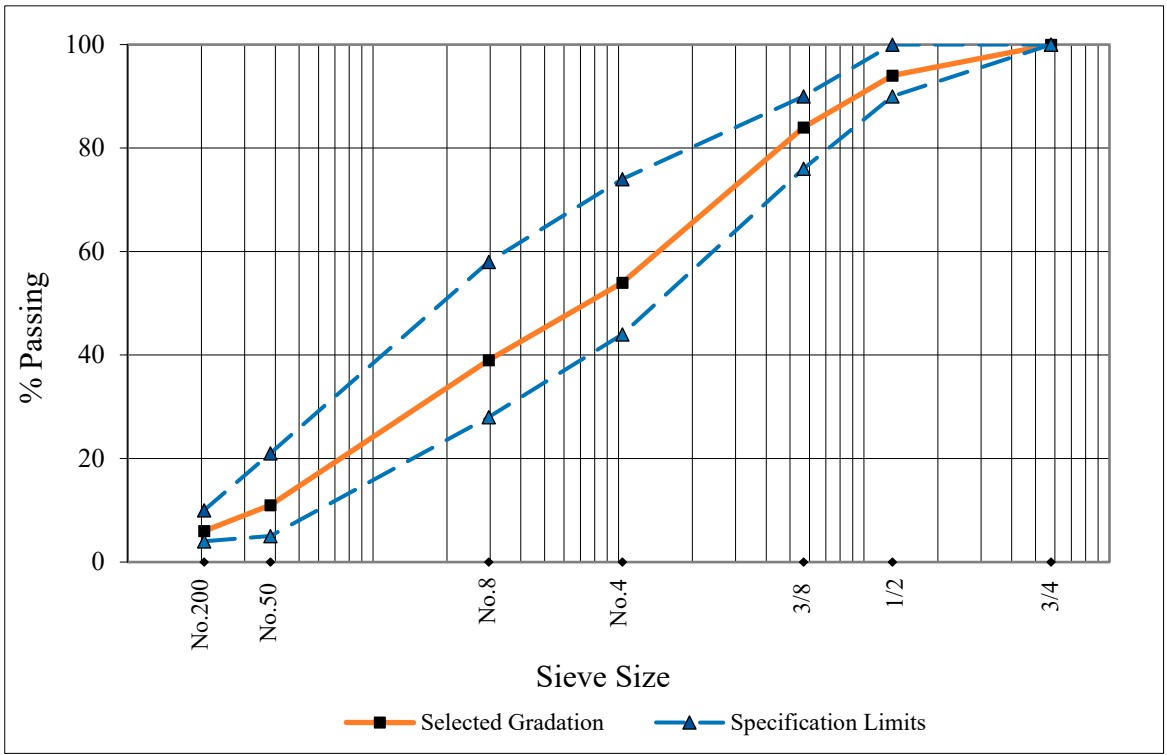

**Figure 2.** The aggregate gradation for surface course.

**Table 3.** Physical properties of coarse aggregate for binder layer.

| Property | ASTM Designation | Test Results | Specification Requirements |
|---|---|---|---|
| Bulk Specific Gravity | ASTM C-127 | 2.564 | . . . . |
| Apparent Specific Gravity | ASTM C-127 | 2.656 | . . . . |
| Percent Water Absorption | ASTM C-127 | 0.424 | . . . . |
| Percent Wear, Los Angles Abrasion | ASTM C-131 | 18 | Max. 35, Binder |
| Flat and Elongated Ratio (5:1) | ASTM D-4791 | 4 | Max. 10 |
| Percent Fractured Pieces | ASTM D-5821 | 94 | Min. 90 |
| Soundness, 5 cycles, $MgSO_4$ | ASTM C-88 | 3.83 | Max. 18% |

**Table 4.** Physical properties of coarse aggregate for surface layer.

| Property | ASTM Designation | Test Results | Specification Requirements |
|---|---|---|---|
| Bulk Specific Gravity | ASTM C-127 | 2.593 | . . . . |
| Apparent Specific Gravity | ASTM C-127 | 2.679 | . . . . |
| Percent Water Absorption | ASTM C-127 | 0.396 | . . . . |
| Percent Wear, Los Angles Abrasion | ASTM C-131 | 18 | Max. 30, Wearing |
| Flat and Elongated Ratio (5:1) | ASTM D-4791 | 3 | Max. 10 |
| Percent Fractured Pieces | ASTM D-5821 | 96 | Min. 90 |
| Soundness, 5 cycles, $MgSO_4$ | ASTM C-88 | 3.83 | Max. 18% |

**Table 5.** Physical properties of fine aggregate.

| Property | ASTM Designation | Test Results | | Specification Requirements |
|---|---|---|---|---|
| Bulk Specific Gravity | ASTM C-128 | 2.684 | | . . . . |
| Apparent Specific Gravity | ASTM C-128 | 2.727 | | . . . . |
| Percent Water Absorption | ASTM C-128 | 0.706 | | . . . . |
| Plasticity Index | AASHTO T89 | NP | | Max. 4% |
| Percent Deleterious material | AASHTO T112 | 0.63 | | Max. 3% |
| Sand Equivalent | AASHTO T176 | natural | 58 | Min. 45 |
| | | crusher | 72 | |

### 2.2. Asphalt Cement

The asphalt cement was supplied by Al-Dura Refinery. Table 6 lists the physical properties of the asphalt cement.

**Table 6.** Asphalt cement test results.

| Property | ASTM Designation | Test Result | SCRB Specification |
|---|---|---|---|
| Penetration at 25 °C, 100 g, 5 s (0.1 mm) | ASTM D-5 | 47 | (40–50) |
| Ductility at 25 °C, 5 cm/min. (cm) | ASTM D-113 | 145 | >100 |
| Flashpoint (Cleveland open cup), (°C) | ASTM D-92 | 321 | Min. 232 |
| The softening point, (°C) | ASTM D-36 | 56 | ——— |
| Viscosity @ 135 °C, cP | ASTM D-4402 | 650 | Min. 400 |
| Viscosity @ 165 °C, cP | ASTM D-4402 | 145 | ——— |
| Specific gravity at 25 °C | ASTM D-70 | 1.03 | ——— |

### 2.3. Mineral Filler

The filler material used was ordinary Portland cement. Its physical properties are listed in Table 7.

**Table 7.** Portland cement physical properties.

| Property | Result |
|---|---|
| Bulk specific gravity | 3.15 |
| Passing Sieve No. 200 (0.075 mm) | 95% |

### 2.4. Tack Coat Materials

Three types of locally available tack coat materials were selected for the research. They were:

- RC-70 cut back to asphalt.
- RC-250 cut back to asphalt.
- CSS-1h emulsified asphalt.

Tables 8 and 9 list out their physical properties against the specification limits.

**Table 8.** Tack coat test results and specifications.

| Property | | Test Results | | Specification Limits | |
|---|---|---|---|---|---|
| | | **RC-70** | **RC-250** | **RC-70** | **RC-250** |
| Kinematic Viscosity at 60 °C (CSt) | | 102 | 458 | 70–140 | 250–500 |
| Test on residue from the distillation | Viscosity at 60 °C (poise) | 72 | 94 | 60–240 | 60–240 |
| | Ductility at 25 °C (cm) | 110 | 110 | Min 100 | Min 100 |
| Residue from distillation to 360 °C (%) | | 57 | 66 | Min 55 | Min 65 |
| Flash point (tag open cup), (°C) | | 62 | 68 | - | Min 27 |
| Water (%) | | Nil | Nil | Max. 0.2 | Max. 0.2 |
| Residue solubility in trichloroethylene (%) | | 99.3 | 99.3 | Min 99.0 | Min 99.0 |

**Table 9.** Emulsified asphalt test results and specifications.

| Property | Test result | Specification Limits | |
|---|---|---|---|
| | | **Min.** | **Max.** |
| Viscosity, Saybolt-Furol at 25 °C | 26 | 20 | 100 |
| Residue by distillation (%) | 58.3 | 57 | - |
| Residue by evaporation | 54.9 | 50 | - |
| Sieve test, (%) | 0.02 | - | 0.0 |
| Cement mixing test,% | 0.732 | - | 2.0 |
| Settlement test, 5-day,% | 0.1 | 0 | 1 |
| One-day storage stability test,% | 0.04 | 0 | 1 |
| Penetration, 25 °C, 100 g, 5 s | 133 | 100 | 250 |
| Ductility, 25 °C cm/min | 185 | 40 | - |
| Solubility in trichloroethylene,% | 99 | 97.5 | - |
| Specific gravity at 25 °C | 1.02 | - | - |

## 3. Mix Design and Production

Marshal tests were conducted to design the HMA mixes in terms of the stability, flow, air voids, and density as per the ASTM D6927 [23]. A total of thirty specimens for both the binding and surface layers were cast in cylindrical molds of a diameter of 4 in and a height of 2.5 in. Figure 3 shows the Marshall test device.

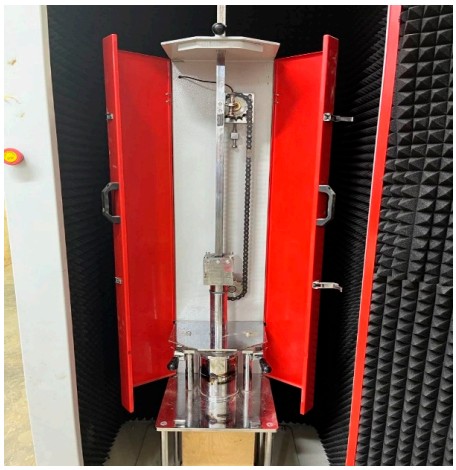

**Figure 3.** Marshall test device.

Figures 4 and 5 show the Marshall test results for the binding layer and the surface layer mixes. Based on the results, it was decided that the optimum asphalt content for the final mix design would be 4.5% for the binding layer mix and 4.8% for the surface layer mix.

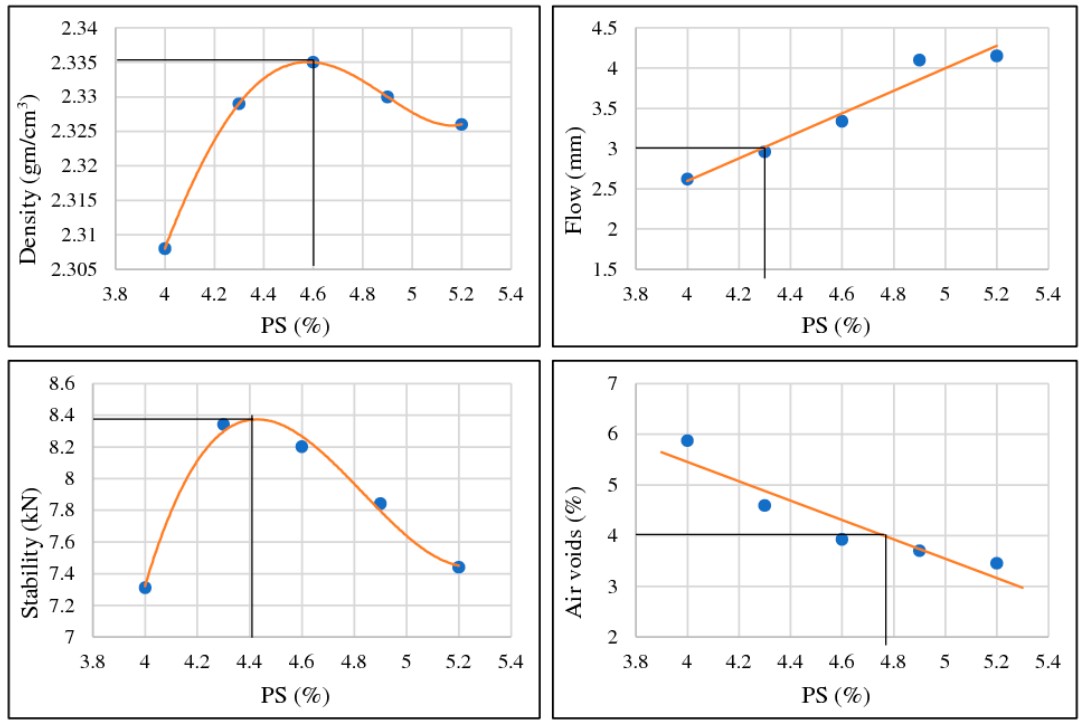

**Figure 4.** Relationship of density, flow, stability, and air voids (%) versus asphalt content (PS) (%) for binder layer.

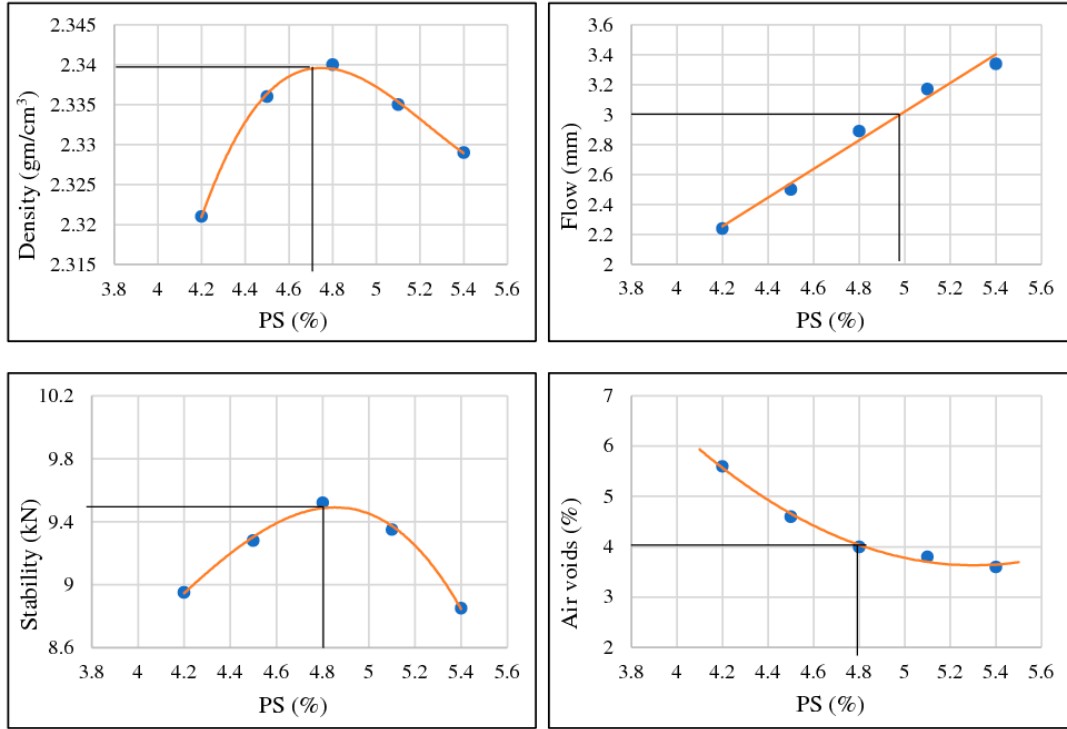

**Figure 5.** Relationship of density, flow, stability, and air voids (%) versus asphalt content (PS) (%) for the surface layer.

All the designed HMA mixes confirmed by the Marshall test were made at an asphalt patch plant using Linnhoff Compact Mix with a production capacity of 120 ton/h. The mixes were produced, respectively, by heating the aggregate in the drying drum up to 150 °C and the asphalt cement in the tank up to 155 °C. Thereafter, they were mixed together at 155 °C. The produced HMA mixes were unloaded into a hopper and dispatched in situ and using a truck to the laboratory, as shown in Figure 6.

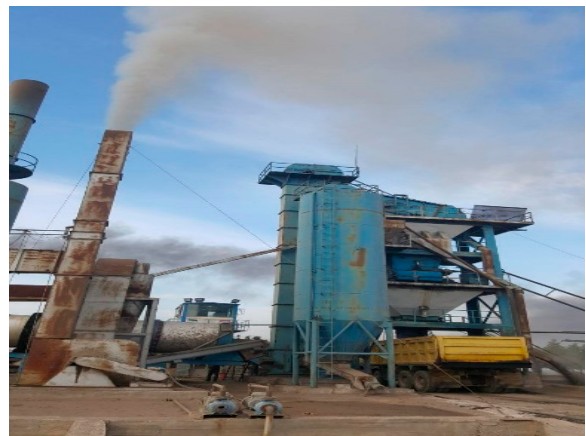 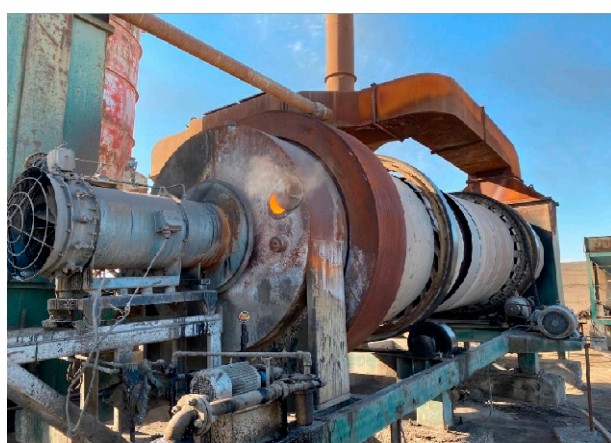

**Figure 6.** HMA production in mix plant.

## 4. Laboratory Test

### 4.1. Specimens Preparation

After the arrival of the HMA mixes at the laboratory, they were cast into rectangular slabs in molds with the dimensions of 300 × 300 × 160 mm, as shown in Figure 6. The molds were preheated to a temperature of 150 °C beforehand; then, the HMA mix for the binding layer was laid into the molds at a thickness of up to 80 mm and compacted by an 8-ton load force using a hydraulic compressor in accordance with an SCRB procedure [19].

Thereafter, the compacted binding course mix was left to cool to 30 °C; then, the preheated tack coat materials, RC-70 and RC250, at a temperature of 60 °C, were spread on the surface of the binding course mix using a plastic brush. The application rates for each type of tack coat were 0.1, 0.2, 0.3, 0.4, and 0.5 lt./m$^2$, respectively. The reason a plastic brush was used was to ensure that there was no absorption of the tack coat material onto the brush. The applied tack coats on the binding layer mix were left to cure for 15 min before the HMA mix for the surface layer was added into the molds and laid on the top surface of the tack coats. The surface layer mix was added up to a thickness of 50 mm in the molds.

After the addition of the surface layer mix, all the materials in the molds were compacted again using the same 8-ton load force. Thereafter, the completed pavement samples were left in the molds for 24 h at room temperature. Finally, cylindrical cores, the test specimens, were taken from the produced HMA concrete slabs using drilling. A total of four core specimens were taken and tested for each type of tack coat at each application rate. A pavement sample with no tack coat was also prepared for the purpose of comparison. Figure 7 shows the procedure for making the samples, the drilling, and the extracted cylindrical specimens.

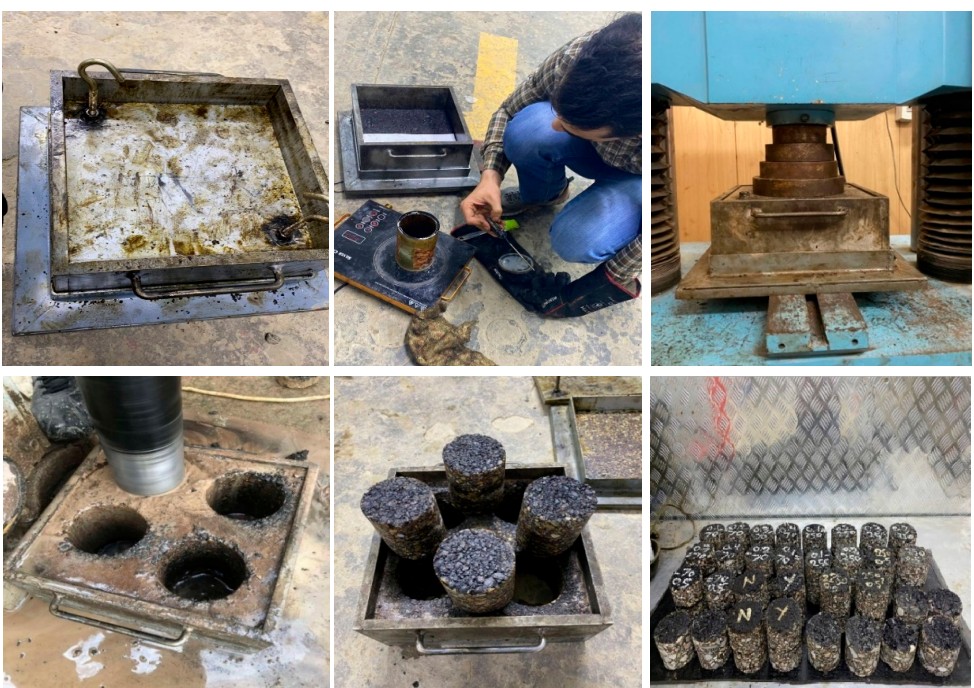

**Figure 7.** The pavement sample preparation and the specimens for shear test.

*4.2. The Bond Strength Evaluation*

The direct shear test has been recognized as one of the primary methods in laboratories for testing the interface bond strength due to its straightforward simplicity. The testing device consists of two parts; one is fixed and the other one is movable, as illustrated in Figure 8. The specimens were tested by putting the end of the binder layer into the fixed part, a ring holder, while the other end of the surface layer was subjected to a cross-section force, a direct shear load, which was applied by the movable part; the force was applied at a position 10 mm away from the joint surface between the two layers. The direct shear load was applied at a rate of 50.8 mm/minute. The bond shear strength was calculated by the division of the highest applied load by the cross-section area of the specimens. Three replicates were used for each test variable and the average result was recorded. The selected test temperature was 30 °C, which represents an intermediate temperature for an in-service pavement.

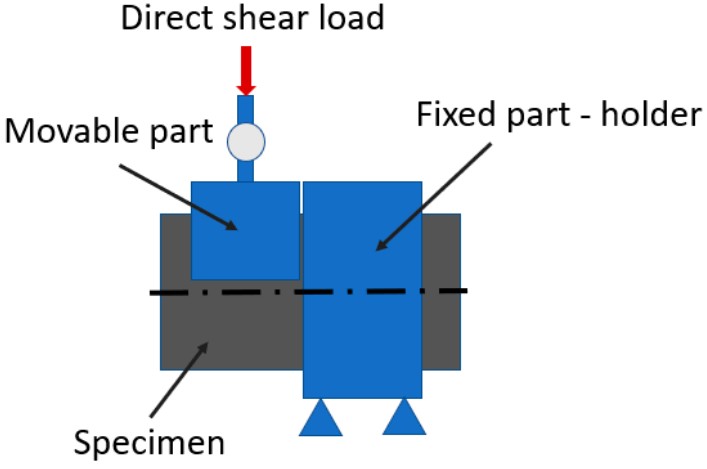

(**a**) Experiment setup

**Figure 8.** *Cont.*

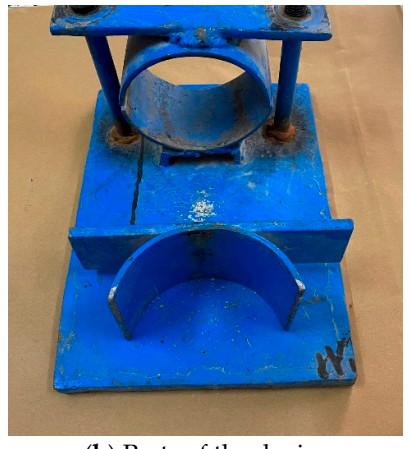
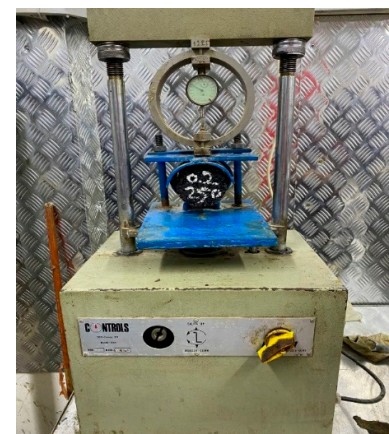

|  |  |
|:-:|:-:|
| (**b**) Parts of the device | (**c**) Direct shear test |

**Figure 8.** Direct shear test in laboratory.

### 4.3. Test Results

The bond strength results, as worked out from all the tests, are given out in Table 10 and are plotted out in Figure 9 for visual comparison.

**Table 10.** Bond strength results for different tack coat types and application rates.

| Application Rate, L/m² | 0 | 0.1 | 0.2 | 0.3 | 0.4 | 0.5 |
|:-:|:-:|:-:|:-:|:-:|:-:|:-:|
| **Tack Coat Type** | **Bond Strength at Various Application Rates (psi)** | | | | | |
| RC-70 | 69.2 | 75 | 62 | 39 | 24 | 18.16 |
| RC-250 | 69.2 | 86 | 90 | 77 | 51.5 | 43 |
| CSS-1 | 69.2 | 78 | 82 | 85 | 64 | 41 |

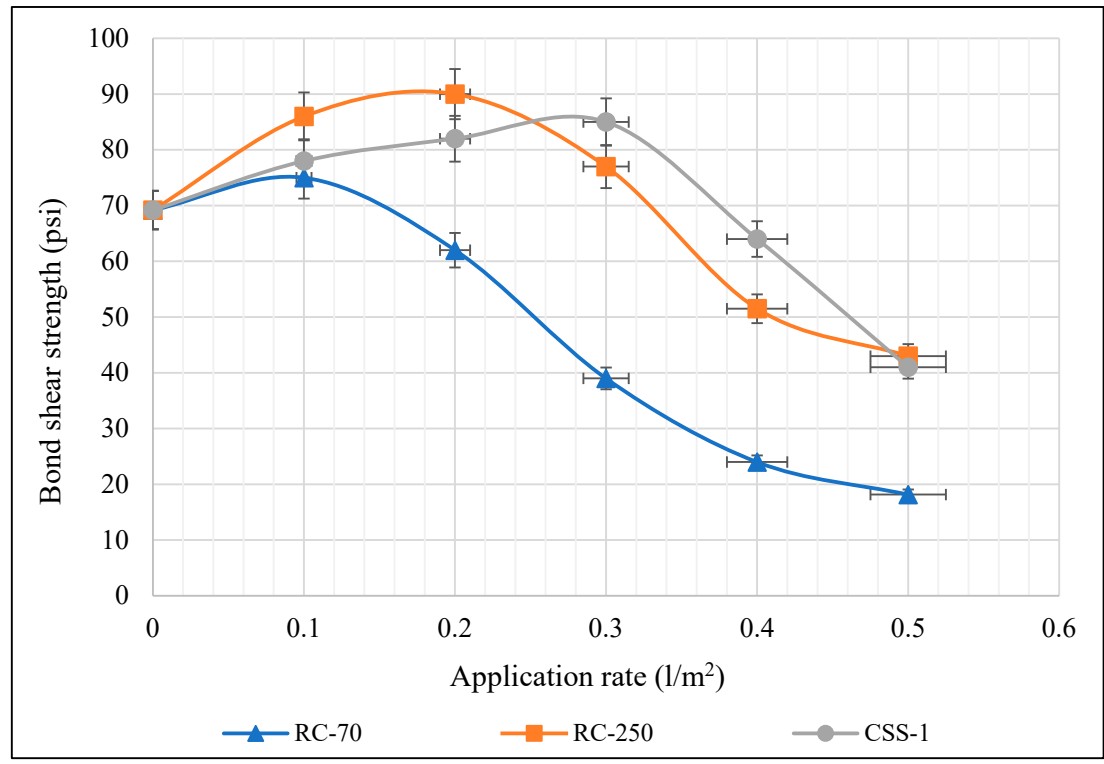

**Figure 9.** Bond strength results for different tack coat types and application rates.

From Figure 8, it can be seen that the optimum application rates are 0.1 L/m$^2$ for RC-70, 0.2 L/m$^2$ for RC-250, and 0.3 L/m$^2$ for CSS-1, respectively. For the three types of tack coat, the RC-250 shows the best binding strength, which is 90 psi. The ratios of the strengths to the application rates are 750, 450, and 283.3 for RC-70, RC-250, and CSS-1, respectively.

## 5. Field Test

The field test was conducted in line with the laboratory study to compare the effect of the tack coat RC-250. The field-testing site is on the Baghdad–Kut highway, located in the southeast of Baghdad city. The HMA mixes produced in the patch plant were delivered to the site and placed by the spreader machine to construct the pavement.

After paving the binding layer course, first the steel plates with 500 × 500 mm dimensions were weighed and placed on the surface of the binder course at a distance of 1.3 m from the edge of the road, with a 2 m distance between each plate. Then, the tack coat spraying vehicle applied the RC-250 tack coat. The vehicle moved at different speeds to ensure different application rates. After the application of the tack coat, the plates were removed and the locations of the plates were marked. The plates were weighed with the tack coat to calculate the tack coat application rate. After the placement of the surface layer, cores were taken from the marked areas. Figure 10 shows the field test stages in the selected locations.

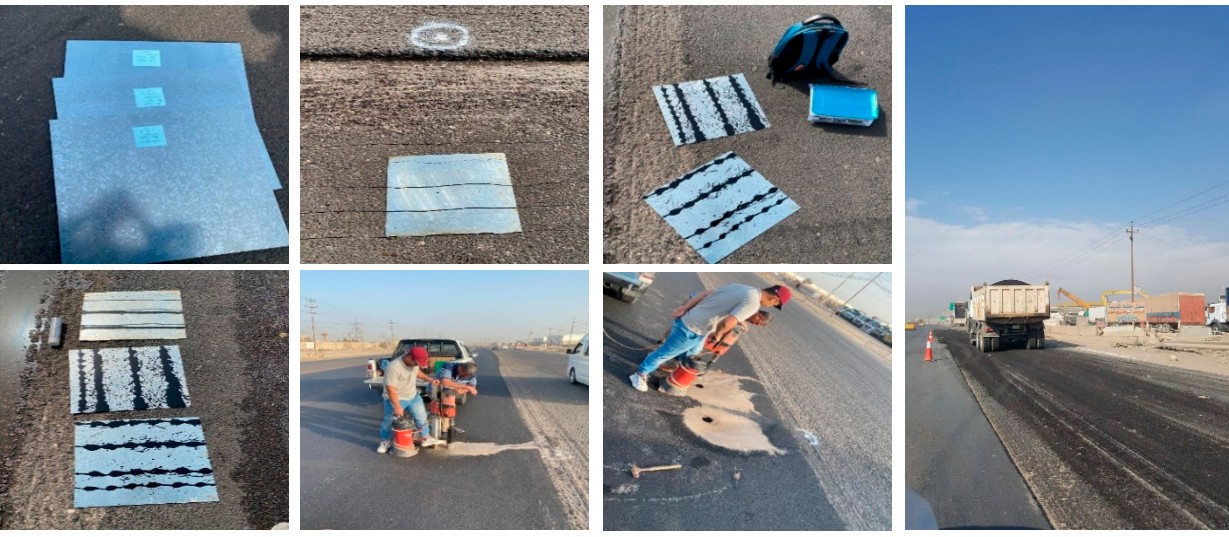

**Figure 10.** Field test stages.

The cores were tested in the direct shear device to measure the bond strength at different application rates; the results are shown in Table 11. Figure 11 shows the direct shear test for the core specimen taken from the construction site.

**Table 11.** Bond shear strength results from field experiment.

| Application Rate, L/m$^2$ | 0 | 0.1 | 0.2 | 0.3 | 0.4 | 0.5 |
|---|---|---|---|---|---|---|
| **Tack Coat Type** | \multicolumn{6}{c}{Bond Shear Strength at Various Application Rates (psi)} | | | | | |
| RC-250 | 61 | 69 | 74 | 59 | 44 | 37 |

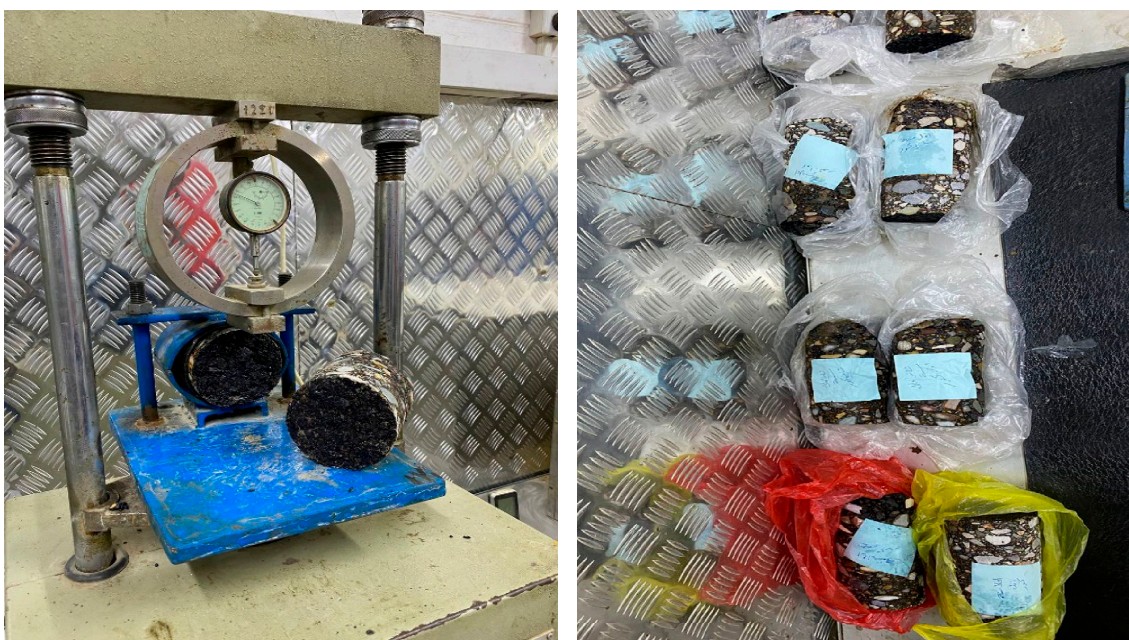

**Figure 11.** Direct shear test for field experiment samples.

After measuring the interface shear strength for the RC-250 tack coat from the lab-prepared specimens as well as the field specimens, a comparison was made, as shown in Figure 12. The results showed that both the laboratory and the field specimens had the same trend and optimum application rate, which was found to be 0.2 L/m$^2$. The interface bond strength measured for the laboratory-prepared specimen was higher than the field core specimen. This variation in the results could be attributed to several factors, such as the degree and type of compaction, the high temperature control of laboratory-prepared specimens, and the difference in surface roughness and cleanliness before the placement of a new layer.

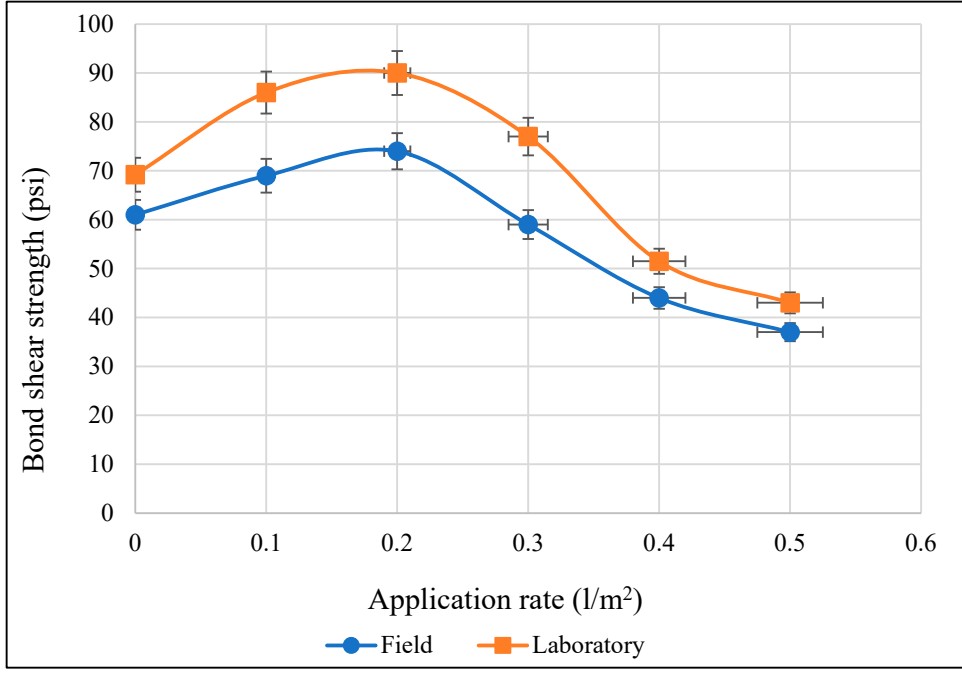

**Figure 12.** A comparison between interface bond strengths for laboratory and field specimens.

## 6. Conclusions

This research examined the interface bond strength between the binder and the surface layers of asphalt concrete pavement for different types and application rates of tack coat materials. The tack coat materials considered were RC-70, RC250, and CSS-1h, with the application rates of (0.1, 0.2, 0.3, 0.4, and 0.5 L/m$^2$) for each one. The selection of the tack coat type complied with the specification requirement as well as the availability of the tack coat material in local markets.

In general, the results from the laboratory test showed that the RC-250 tack coat gave a higher interface bond strength (90 psi) when compared with the other types used in this study; moreover, the optimum application rate was 0.2 L/m$^2$, whereas the RC-70 showed less interface bond shear strength.

The test results for the specimens with RC-70 tack coat material showed a maximum interface bond strength of 75 psi at an optimum application rate of 0.1 L/m$^2$. For CSS-1h, the maximum interface bond strength was found to be 85 psi at a 0.3 L/m$^2$ optimum application rate.

The results from the field-prepared specimen using the RC-250 tack coat at the Baghdad–Kut highway site showed that the maximum interface bond shear strength equaled 74 psi and that the optimum application rate was 0.2 L/m$^2$. This variation in the results could be attributed to several factors, such as the degree and type of compaction, the high temperature control of the laboratory-prepared specimens and the difference in surface roughness and cleanliness before the placement of a new layer.

**Author Contributions:** Conceptualization, A.H.K. and Y.W.; methodology, M.H.A.; validation, M.H.A.; formal analysis, Y.W.; investigation, Y.W. and A.H.K.; resources, A.H.K.; data curation, Y.W.; writing—original draft preparation, M.H.A.; writing—review and editing, M.H.A.; visualization, M.H.A.; supervision, A.H.K.; project administration, A.H.K.; funding acquisition, M.H.A. All authors have read and agreed to the published version of the manuscript.

**Funding:** This research received no external funding.

**Institutional Review Board Statement:** Not applicable.

**Informed Consent Statement:** Not applicable.

**Data Availability Statement:** Not applicable.

**Conflicts of Interest:** The authors declare no conflict of interest.

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
