# Peer review of "Experimental Study of the Effect of Tack Coats on Interlayer Bond Strength of Pavement"

_sustainability, doi:10.3390/su15086600_

Round 1

Reviewer 1 Report

This paper investigates the effect of tack coats on interlayer bond strength of pavement, and the three materials on the bond strength of the pavement was analyzed. The research objectives are clear and demonstrated in the project, so, It is recommended to reconsider after major revision. The main comments are as follows.

(1)The tables in the article are too poorly formatted and not laid out in accordance with MDPI publication requirements.

(2)The introduction section of the article is written separately from the previous studies section of the study; it is recommended to combine the two section and name the title as “Introduction”. Please refer to relevant research papers (doi.org/10.1016/j.jclepro.2022.134123, doi.org/10.1016/j.conbuildmat.2022.130054).

(3)The main purpose of this paper is to study the interfacial bond strength, and it is not recommended to use more content to illustrate the Marshall index of both binding layer mix and surface layer mix. Such as Figures 4 and 5.

(4)How did you ensure that the thickness of the layers was consistent with the design during the experiment? What method did you use? If so, please add to the text.

(5)Please add the number of parallel tests of specimens in the interfacial bond strength.

(6)Please add error lines to Figures 9 and 12.

(7)Please add the test temperature for interfacial bond strength.

(8)Interface bond strength is not only related to the material and thickness, but also should consider the morphology of the asphalt mixture contact surface. How did you consider during the experiment?

Author Response

The authors wish to thank the reviewer for the time and effort in reviewing our manuscript. The provided comments and constructive suggestions are very helpful in improving the quality of our manuscript and have helped us identify several shortcomings in our manuscript. All the provided comments were addressed.

Reviewer 2 Report

The paper examines the interface bond strength between the binder and the surface layers of asphalt concrete pavement. It investigates various types of tack coat and determines the optimal application rate for each type. The paper is well organized; but requires significant technical editing. The followings are suggested comments to improve the manuscript quality:

1. It seems the information in Figure 1 and Figure 2 has been covered Table 1 and Table 2. Please remove the Figure 1 and Figure 2 if that is the case.

2. This study covers three tack coat types: RC-70, RC-250, and CSS-1h. It is recommended to provide more information on why they are selected in this study. Are they representative enough to all the tack coat types in the market?

3. Section 3 lists the properties of all the materials used in this study. Instead of trivially presenting all information, it is recommended to organize and summarize the material properties. Can we interpret from this section that the binder layer and surface layer use common materials?

4. The laboratory tests are performed on all the three tack coat types. However, the field test provides only the field performance of tack coat RC-250. Although we can see from Figure 12 that the field test results align well with the laboratory test results, it is not clear if this statement hold true for other two tack coat types.

5. How is the application rate measured? Figure 9 shows the application rate as a percentage. But Figure 12 shows it as l/m2.

6. Inadequate bonding between layers can result in some long-term performance issue such as path cracking, fatigue cracking, potholes, and rutting. Is there any long-term monitor plan to identify the impact of tack coating on the pavement life?

7. Too much tack coat can create a lubricated slippage plane between layers, or can cause the tack coat material to be drawn into an overlay, negatively affecting mix properties and even creating a potential for bleeding in thin overlays. It is recommended to discuss if the application rate of the tack coat could affect other mixture performance apart from bond shear strength.

Author Response

(The authors gave the same response as above.)

Round 2

Reviewer 1 Report

All my comments are addressed, I am okay with the publish of this work.

Author Response

The authors wish to thank the reviewer for the time and effort in reviewing our manuscript

Reviewer 2 Report

Some of the comments are not properly addressed. Please respond to them to improve the quality of this paper:

1. The information in Figure 1 and Figure 2 has been covered in Table 1 and Table 2. Please take down the Figure 1 and Figure 2 if they do not provide additional information.

2. Although the three coat types are the only types available in the local market, it is suggested to look at more coat types, especially the asphalt emulsions.

3. The author did not provide a proper answer to this question:

Section 3 lists the properties of all the materials used in this study. Instead of trivially presenting all information, it is recommended to organize and summarize the material properties. Can we interpret from this section that the binder layer and surface layer use common materials?

4. The author did not provide a proper answer to this question:

The laboratory tests are performed on all the three tack coat types. However, the field test provides only the field performance of tack coat RC-250. Although we can see from Figure 12 that the field test results align well with the laboratory test results, it is not clear if this statement hold true for other two tack coat types.

5. The author need to answer how is the application rate measured? 

6. Inadequate bonding between layers can result in some long-term performance issue such as path cracking, fatigue cracking, potholes, and rutting. Is there any long-term monitor plan to identify the impact of tack coating on the pavement life? Please discuss in the paper

7. Too much tack coat can create a lubricated slippage plane between layers, or can cause the tack coat material to be drawn into an overlay, negatively affecting mix properties and even creating a potential for bleeding in thin overlays. It is recommended to discuss if the application rate of the tack coat could affect other mixture performance apart from bond shear strength. Please discuss in the paper

Author Response

(The authors gave the same response as above.)

Round 3

Reviewer 2 Report

All my comments have been well address. Thanks to authors for their work.